# Promising Results About the Possibility to Identify Prostate Cancer Patients Employing a Random Forest Classifier: A Preliminary Study Preoperative Patients Selection

**DOI:** 10.3390/diagnostics15040421

**Published:** 2025-02-10

**Authors:** Eliodoro Faiella, Matteo Pileri, Raffaele Ragone, Anna Maria De Nicola, Bruno Beomonte Zobel, Rosario Francesco Grasso, Domiziana Santucci

**Affiliations:** 1Unit of Radiology and Interventional Radiology, Fondazione Policlinico Universitario Campus Bio-Medico, Via Alvaro del Portillo, 00128 Rome, Italy; e.faiella@policlinicocampus.it (E.F.); matteo.pileri@unicampus.it (M.P.); annamaria.denicola@unicampus.it (A.M.D.N.); b.zobel@policlinicocampus.it (B.B.Z.); r.grasso@policlinicocampus.it (R.F.G.); d.santucci@policlinicocampus.it (D.S.); 2Research Unit of Radiology, Department of Medicine and Surgery, Università Campus Bio-Medico di Roma, Via Alvaro del Portillo, 00128 Rome, Italy

**Keywords:** prostate cancer, lymph node involvement, multiparametric MRI, artificial intelligence, radiomics, Random Forest model

## Abstract

**Objective**: This study evaluates the accuracy of a Machine Learning model of Random Forest (RF) type, using MRI data and radiomic features to predict lymph node involvement in prostate cancer (PCa). **Methods**: Ninety-five patients who underwent mp-MRI, prostatectomy, and lymphadenectomy at the Fondazione Policlinico Campus Bio-medico Radiological Department from 2016 to 2022 were analyzed. Radiomic features were extracted from T2-weighted, DWI, and ADC sequences and processed using a Random Forest (RF) model. Clinical data such as PSA levels and Gleason scores were also considered. **Results**: The RF model demonstrated significant accuracy in predicting lymph node involvement, achieving 84% accuracy for nodules in the peripheral zone (80% for predicting positive lymph node involvement and 85% for negative lymph node involvement) and 87% for those in the transitional zone (86% for predicting positive lymph node involvement and 88% for negative lymph node involvement). In the peripheral zone, key features included ADC shape maximum 2D diameter row and T2 noduloglcm difference variance, while in the transitional zone, DWI glcm difference average and DWI glcm Idm were important. DWI and ADC sequences were particularly crucial for accurate lymph node assessment. First-order features emerged as the most significant in whole-gland analysis, indicating fundamental differences in tumor composition and density critical for identifying malignancies with higher metastatic potential. **Conclusions**: AI-driven radiomic analysis, especially using DWI- and ADC-derived features, effectively predicts lymph node involvement in PCa patients, in particular in negative linfonode status patients, offering a promising tool for preoperative linfonode sparing patient selection. Further validation with larger cohorts is needed. Some limitations of this study are a relatively small sample size and it being a retrospective study.

## 1. Introduction

Prostate cancer (PCa) is among the most prevalent malignancies and the second leading cause of cancer-related deaths in males. In 2023, Europe witnessed 1,261,990 cases of cancer-related deaths [1]. The identification of pelvic lymph node metastasis (PLNM), present in approximately 15% of newly diagnosed prostate cancer patients, is a crucial prognostic factor that correlates with biochemical recurrence and distant metastases after curative treatment. Therefore, precise pre-treatment detection of PLNM in localized PCa cases could dramatically influence clinical decision-making processes, the formulation of treatment strategies, and the prediction of patient outcomes [2].

Advancements in artificial intelligence (AI) have propelled radiomics to the forefront of innovation, concentrating on the extraction of quantitative imaging features that non-invasively predict the behavior of nodules and tumors. This approach has the potential to overcome certain limitations in diagnostic accuracy associated with human interpretation. Radiomics has recently emerged as a critical tool for providing a more quantitative and objective evaluation of medical imaging. Through the analysis of quantitative attributes of tumor intensity and morphology, radiomic features can potentially reveal intratumoral histopathological characteristics, thus offering valuable prognostic insights for cancer management [3]. Among different ML algorithms, Random Forest has demonstrated a high effectiveness for handling high-dimensional datasets due to several advantages such as feature subset selection, dimensionality reduction without explicit feature elimination, resistance to overfitting, and parallel processing.

Magnetic Resonance Imaging (MRI) has been extensively adopted in radiomic analyses, particularly for PCa, owing to its ability to extract advanced features from images with high contrast and various weighting types. In radiomics studies, there are several type of radiomics features that can be taken out from a MRI image, based on the purpose of the specific study: the most common used are intensity, texture, shape of a region of interest, and gray intensity, in different types of weighted image [4]. The prostate is distinguishable into two regions on MRI scans: the peripheral zone (PZ), which exhibits hyperintensity on T2-weighted MRI owing to its glandular-ductal tissue, and the transition zone (TZ), which comprises stromal tissue and appears hypointense on T2-weighted images. Additionally, DWI/ADC sequences are crucial for delineating target areas, highlighted by the hyperintensity/hypointensity of suspicious nodules. Both T2WI and DWI/ADC sequences offer numerous features that can be leveraged to extract a variety of information.

The aim of this study is to evaluate the accuracy of an AI model of RF type in predicting lymph node involvement in prostate cancer using radiomic features from MRI sequences. The study also seeks to identify the most informative MRI sequences and radiomic features that contribute to accurate predictions, enhancing preoperative assessment and potentially reducing unnecessary invasive procedures.

## 2. Materials and Methods

This retrospective study analyzed 95 patients who underwent multiparametric magnetic resonance imaging (mp-MRI) followed by prostatectomy and lymphadenectomy at the Radiological Department of our hospital between 2016 and 2022. Inclusion criteria mandated that patients had both procedures. Clinical data collected included age, pre-mp-MRI PSA levels, MRI characteristics (signal intensity on T2-weighted images, DWI/ADC maps, and PI-RADS score), and histological tumor details (Gleason Score, TNM staging, capsule invasion, seminal vesicle invasion, neurovascular bundle involvement, and histological type). The cohort was divided into two groups based on lymph node status: 35 patients with at least one positive lymph node (positive group) and 60 patients with no lymph node metastasis (negative group).

The research protocol adhered to the ethical standards laid down by current legislation and the Declaration of Helsinki. The need for ethical review and approval was waived owing to the retrospective nature of this study. All patients signed an informed consensus before MRI execution.

### 2.1. Magnetic Resonance Imaging

MRI scans were performed using a 1.5 Tesla Magnetom Aera scanner (Siemens^®^, Erlangen, Germany). The mp-MRI protocol included multiplanar T2-weighted images, echo-planar DWI with b-values of 0, 800, or 1000 s/mm^2^ (ADC maps were automatically calculated), dynamic contrast-enhanced imaging (DCE), and axial T1-weighted images.

Patients were scanned in a supine position using an external coil, covering the prostate gland and seminal vesicles from the pelvis to the aortic bifurcation. Preoperative mp-MRI was performed systematically 4–5 weeks post-ultrasound-guided transrectal biopsy.

### 2.2. Segmentation, Feature Extraction and Selection

All mp-MRI data were transferred to a dedicated workstation. Manual segmentation of prostate nodules was performed on T2-weighted sequences, DWI, and ADC maps using 3D Slicer (v. 5.0.1). The segmentations were performed using a dedicated tool (“segment editor”), creating a mask of the region of interest (ROI) on the three different dimensional planes (axial, sagittal and coronal), and obtained a volume of interest (VOI). As we can see in Figure 1, the ROI is represented by a green zone on the images. Three volumes of interest (VOIs) were extracted per patient, yielding 339 radiomic features per patient after analysis with PyRadiomics software (version 3.0). Features included first-order statistics, shape-based features, and texture features such as those from the Gray-Level Co-occurrence Matrix (GLCM), Gray Level Dependence Matrix (GLDM), Gray-Level Run-Length Matrix (GLRLM), Gray-Level Size Zone Matrix (GLSZM), and Gray-Level Distance Zone Matrix (GLDZM).

In particular:-Shape features: these characterize the spatial attributes of the region of interest (ROI), including its surface area, overall volume, diameter, elongation, sphericity, and the ratio of surface area to volume.-First-order statistics (histogram-based features): these parameters describe the distribution of voxel intensities within the image region of interest, utilizing established statistical measures such as energy, entropy, mean, interquartile range, skewness, kurtosis, and uniformity.-Second-order statistics (textural features): these methods analyze the statistical relationships between adjacent voxels. The significant features include the following:○Gray-level Cooccurrence Matrix (GLCM): examines the three-dimensional arrangement of grayscale intensity values within a volumetric image.○Gray-Level Run-length Matrix (GLRLM): evaluates adjacent voxels sharing identical gray-level values, assessing the length of gray-level runs in multiple orientations.○Gray-Level Size-Zone Matrix (GLSZM): measures the regions of adjacent voxels with identical gray-level intensity within a three-dimensional image.○Neighboring Gray-Tone Difference Matrix (NGTMD): determines the contrast between a voxel’s intensity and the mean intensity of surrounding voxels within a defined radius.○Gray-level Dependence Matrix (GLDM): evaluates the quantity of interconnected voxels within a specified range that are reliant on the central voxel.

Coefficient value for all selected features was calculated considering the Random Forest model (RF importance) both for positive and negative prediction and calculated for both the peripheral and transitional prostatic zone and considering all features. RF importance value (based on Gini impurity) was reported for T2 sequences, ADC, and DWI, and was considered for the 5 most- and least-significant features. For the analysis, we just used all features.

### 2.3. Radiomics Analysis, Random Forest Model Development, and Features Identification

The study utilized a Random Forest (RF) algorithm for its efficacy in handling high-dimensional datasets, as we demonstrated in our previous article [5] through the comparison with two other ML classifiers. Separate RF models were constructed for each imaging sequence (DWI, ADC, and T2-weighted images). The model’s performance was evaluated using metrics such as accuracy, precision, recall, F1-Score, and Area Under the Receiver Operating Characteristic Curve (AUC). The most impactful features within the AI models were identified and their RF importance values reported, particularly for T2 sequences, ADC, and DWI. ChatGPT version 4.0 was used for statistical analysis [6].

In the Random Forest model, feature importance is a measure of how much each variable contributes to improving the model’s ability to make accurate predictions. To calculate feature importance in Random Forest, we used the Gini impurity reduction method, which is one of the most common techniques for assessing how much a feature contributes to improving the model.

Before using the Gini method, the dataset was split to divide the dataset into a training set and a validation set. The training set is used to train the model, while the validation set is used to test its performance during training, allowing for monitoring of potential overfitting.

So, it then continues with Gini method. It is as follows: a feature is used to make a split in the data within a decision tree, and it calculates how much this split reduces the impurity in the node. Gini impurity measures how mixed the classes are within a group: If a group contains mostly data from one class (e.g., patients with or without lymph node invasion), the impurity is low. If the group is highly mixed (containing both patients with and without lymph node invasion), the impurity is high.

The goal of each decision tree is to reduce the impurity at each node. The feature that causes the greatest reduction in impurity, by better separating the classes, is considered the most important. This process is repeated for all the trees in the Random Forest, and the final importance of each feature is calculated by summing the total reduction in impurity caused by that feature across all the trees. The features that contribute most to reducing impurity are those that the model considers most significant for making accurate predictions.

The selected features are the ones that led to the best results in the Random Forest model in terms of accuracy, precision, recall. The selection process, based on improving class separation through Gini impurity reduction, identified these features as the ones that provided the best balance between reducing error and improving model interpretability.

## 3. Results

The study included 95 patients, of whom 35 had metastatic nodules and 60 did not. The cohort consisted entirely of Caucasian individuals with a mean age of 60.3 y, with a median PSA level of 4.5 ng/mL and a median Gleason score of 7. Tumors were predominantly located in the peripheral zone (63 patients) and less frequently in the transitional zone (32 patients) (Table 1).

A single RF model was created considering only patients with prostate cancer. Subsequently, the same model was applied exclusively to patients with prostate cancer in the transition zone (TZ) and the peripheral zone (PZ). The reported results (in Tables 3–7) correspond to the validation subset. The model exhibited high overall accuracy but showed a notable number of false positives (80) and false negatives (20)**.** This indicates a tendency to favor the majority class, which could explain the lower AUC despite strong overall classification performance (Table 2).

The Random Forest (RF) model demonstrated significant accuracy in predicting lymph node involvement, achieving 84% accuracy for nodules in the peripheral zone (PZ) and 87% for those in the transitional zone (TZ) (Table 3). Specifically, in the PZ, the accuracy for predicting positive lymph node involvement (PL) was 80%, and for negative lymph node involvement (NL) it was 85%. In the TZ, the accuracy for PL was 86%, and for NL it was 88% (Table 4).

The radiomic feature analysis of the study resulted in the extraction of three volumes of interest (VOIs) per patient—derived from the tumor segmentation on T2 sequences, DWI, and ADC maps. A total of 113 radiomic features were selected from each VOI, yielding 339 features per patient after analysis with the PyRadiomics software and subsequent processing by GPT-4. Key radiomic features significantly influenced the predictive performance of the RF models. In the PZ, the most influential features for positive LN prediction included ADC_shape_Maximum_2D_Diameter_Row and T2_noduloglcm_Difference_Variance, while for negative LN prediction, features like DWI_shape_Maximum_2D_Diameter_Row and ADC_glrlm_Long_Run_High_Gray_Level_Emphasis were crucial (Table 5). In the TZ, important features for positive LN prediction were DWI_glcm_Difference_Average and DWI_glcm_Idm, whereas for negative LN prediction, DWI_glcm_Id and DWI_glcm_Idmn were significant (Table 6). For the entire gland, the top features for positive LN prediction were DWI_firstorder_Total_Energy and DWI_firstorder_Energy, and for negative LN prediction, DWI_glcm_Idmn and DWI_gldm_Large_Dependence_Emphasis were most influential (Table 7).

## 4. Discussion

MRI has become an essential non-invasive modality for assessing lymph node (LN) status, although its effectiveness can be influenced by the expertise of radiologists [7,8]. The T2-weighted sequence plays a crucial role by providing high-resolution anatomical details, particularly useful for delineating the transitional zone and identifying tumor boundaries [9,10]. DWI is a functional imaging method that uses the random translational motion of water molecules to allow noninvasive assessment of biological tissues. Tissue characterization is made possible by the quantitative expression of the degree of diffusion limitation through the computation of ADC maps [9,11,12]. The exclusion of post-contrast dynamic contrast-enhanced (DCE) sequences from this study is based on their use within the PI-RADS framework, where they are primarily recommended for indeterminate cases. This decision aligns with current guidelines, which suggest that DCE sequences provide limited additional diagnostic value in routine assessments [13].

Our study aimed to enhance the prediction of LN involvement by extracting and analyzing radiomic features from MRI sequences and incorporating them into a sophisticated Random Forest (RF) analytical model. The radiomics features were combined with various clinical data points, such as laboratory results, PSA levels, and Gleason scores, to determine the most informative MRI sequence for LN status determination.

A recent review conducted by Huynh et al. highlights the growing body of literature advocating for the incorporation of radiological and radiomic features to improve prognostic models in PCa [14] and demonstrated superior clinical utility in predicting LN involvement, with 33 studies detailing the current scope of MRI-derived radiomics in PCa management. These studies employ various MRI sequences, feature extraction methods, and validation techniques to develop radiomic models with impressive discriminative abilities, as evidenced by the AUC values reported. For instance, a study by Damascelli et al. utilized a combined T2-weighted and ADC model that yielded an AUC of 0.90 for LN involvement prediction, supporting the integrative approach [15]. Another study by Liu et al. demonstrated the predictive strength of machine learning models in determining LN status with AUCs up to 0.86, reinforcing the potential of such technologies in clinical practice [16]. In support of this claim, Hou et al. developed a model based on T2W, DWI/ADC sequences with an AUC of 0.92 for assessing lymph node involvement in internal testing and 0.72 for external testing [17]. However, despite these promising developments in radiomic models for LN prediction in PCa, the labor-intensive process of volume of interest (VOI) acquisition remains a significant barrier to widespread clinical adoption, preventing radiologists from performing radiomic analysis [18].

The radiomic features offer a more profound analysis of prostate cancer that traditional modalities [19].

The radiomic features considered in this study, particularly those derived from DWI and ADC sequences, demonstrated significant predictive power for lymph node involvement in prostate cancer. Our analysis showed that radiomic features were notably more impactful and significant in predicting negative lymph node involvement, especially in the transitional zone. The RF importance scores highlighted that features such as “DWI_glcm_Id” and “DWI_glcm_Idmn” were critical for predicting negative lymph node involvement, with an accuracy reaching 85% in the transitional zone (Table 3). This indicates a higher predictive capability for patients with negative lymph nodes, potentially allowing for the avoidance of unnecessary lymphadenectomies, thus supporting the use of MRI as a non-invasive screening tool.

The features selected and previously extracted from MRI sequences (T2W, DWI, and ADC) were chosen because they are directly linked to the biological and physiological characteristics of the tumor and surrounding tissues, which are crucial for diagnosing and staging prostate cancer. Including these features provides potentially more useful diagnostic information than other features that may be less relevant to lymph node staging.

RF is highly effective for high-dimensional datasets due to its ability to automatically identify relevant features and handle noisy or irrelevant data. Its ensemble approach reduces overfitting, bias, and variance while capturing non-linear relationships without extensive preprocessing. RF is scalable, supports both classification and regression tasks, and handles imbalanced datasets effectively. Its robustness and versatility make it suitable for applications in bioinformatics, finance, and radiomics analysis. The model’s poor performance, particularly in terms of precision and recall for class 0 and class 1, is likely due to the dataset’s imbalance. With only 95 patients and 35 positive cases of lymph node involvement, the model tends to favor the majority class (class 0), leading to missed detections (low recall) and false positives (low precision) for class 1. Additionally, the small dataset increases the risk of overfitting, as the model may rely excessively on specific features that do not generalize well.

In our previous study we explored three algorithms (RF, Logistic regression, and SVM) in order to predict LNS in PCa patients, and we obtained the significantly best performance using RF model [5].

The importance of these radiomic features was particularly pronounced when data from the peripheral and transition zones were analyzed separately rather than collectively. This differentiation underscores the value of individualized zone-specific analysis in enhancing the accuracy of predictive models. The majority of significant features were from the DWI and ADC sequences, not T2-weighted images, reflecting the lesion intrinsic superior contrast resolution of DWI in the transition zone, which makes it preferable for assessing tumor characteristics in this region.

GLCM (Gray-Level Co-occurrence Matrix) features, which capture textural information, were prominently featured among the top predictive indicators. This aligns with findings by Prata et al. who highlighted the critical role of textural features in radiomic models for PCa [20]. Our results show that GLCM features such as “DWI_glcm_Difference_Average” and “DWI_glcm_Id” had high RF importance scores, emphasizing their utility in capturing the intricate textural nuances of tumor heterogeneity that are crucial for accurate lymph node status prediction (Table 4 and Table 5).

Additionally, shape-based radiomic features also played a significant role in the predictive models. Features like “ADC_shape_Maximum_2D_Diameter_Row” and “T2_nodulo_shape_Elongation” were among the most influential for positive lymph node prediction, while “DWI_shape_Maximum_2D_Diameter_Row” was critical for negative lymph node prediction (Table 4). These shape features help characterize the geometric properties of the tumor, providing essential insights into tumor morphology that are complementary to textural features.

When evaluating the predictive value of lymph node involvement for the whole gland, first-order radiomic features emerged as the most significant, particularly in predicting positive lymph node status. This finding suggests that basic statistical measures of voxel intensity within the tumor, such as mean, variance, and skewness, are highly informative. These features likely reflect fundamental differences in tumor composition and density that are critical for identifying malignancies with higher metastatic potential. The prominence of first-order features in whole-gland analysis underscores their utility in capturing overall tumor characteristics that are predictive of positive lymph node involvement.

These findings suggest that radiomic analysis, particularly with DWI and ADC-derived features, can significantly enhance the preoperative assessment of prostate cancer, reducing the need for invasive procedures in patients with negative lymph nodes. This aligns with previous studies that advocate for the integration of radiomic features to improve diagnostic accuracy and prognostic models in prostate cancer management.

Our findings align also with pioneering studies like those of Faiella et al. and Bourbonne et al., which showcased the efficacy of a synergistic clinical-radiomic methodology [19,21]. By condensing an extensive array of features into a focused subset, their work spotlighted the diagnostic significance of T2-weighted sequence features. However, our analysis goes a step further by demonstrating the discriminative strength of DWI features, which not only complement but also enrich the predictive capacity of our model. The incorporation of these DWI features, along with the proven T2-weighted features, echoes the strategy recommended by Huynh et al., championing a combined radiomic–clinical model. This combination not only elevates the predictive accuracy but also weaves a more detailed tapestry of the tumor’s behavioral patterns [14]. By incorporating both DWI and T2-weighted imaging features, we provide a nuanced understanding of the tumor environment, laying down an advanced framework for prognostic modeling. This integrated imaging strategy paves the way for precision medicine, offering enhanced tools for clinical decision-making and personalized patient care strategies, particularly in the predictive modeling of lymph node involvement in prostate cancer.

Zheng et al. [22] developed an integrated radiomics model (IRM) to predict lymph node involvement. This model combined histopathological examination, radiomics features extracted from a prostate index lesion, and clinical characteristics, utilizing only a support vector machine (SVM). The IRM demonstrated strong predictive performance, achieving an area under the curve (AUC) of 0.915 in the testing set. Moreover, the IRM’s AUC significantly outperformed existing clinical nomograms, which ranged from 0.698 to 0.724, with a statistically meaningful difference (*p* ≤ 0.05).

In another research, Liu et al. [16] introduced two preoperative models for evaluation of PLNM, utilizing multivariate logistic regression and incorporating both radiological and radiomics characteristics. These models demonstrated an AUC of 0.89–0.90. Additionally, their research indicated that a DWI-based radiomics nomogram, combining LN radiomics signature with quantitative radiological features, shows potential for predicting PLNM in PCa patients, especially for normal-sized LNM.

Despite promising results, it is important to acknowledge the limitations of our study, including its retrospective nature, modest sample size, and single-institution dataset. These factors may limit the generalizability of our findings and should be taken into consideration when interpreting our results. Future research should aim to validate our findings in a larger and more diverse population to reinforce the utility of this model in clinical practice, also considering including DCE sequences to potentially enhance predictive models for lymph node involvement in PCa.

## 5. Conclusions

This study underscores the potential of AI-driven radiomic analysis for improving the prediction of lymph node involvement in prostate cancer, especially when leveraging DWI and ADC sequences. The radiomic model demonstrated high accuracy, particularly for negative lymph node predictions in the peripheral zone, suggesting that MRI-based radiomics could serve as an effective screening tool to reduce unnecessary invasive procedures. Shape-based features, along with textural features, were crucial in the predictive models, highlighting the multifaceted nature of tumor characterization necessary for accurate lymph node assessment. Further validation with larger, multi-institutional cohorts is necessary to confirm these findings and enhance clinical decision-making and personalized treatment strategies in prostate cancer management. Perspective strategies include higher patient sample, multi-cohort study design, possibly with patients differing on clinic-anamnestic characteristics and employing different MR scans (3T vs. 1.5T) and deep learning techniques application, and longitudinal studies assessment in order to assess the impact of predictive models on clinical outcomes. Even more validation with a larger dataset and open-source sharing would be appreciated in the future.

## Figures and Tables

**Figure 1 diagnostics-15-00421-f001:**
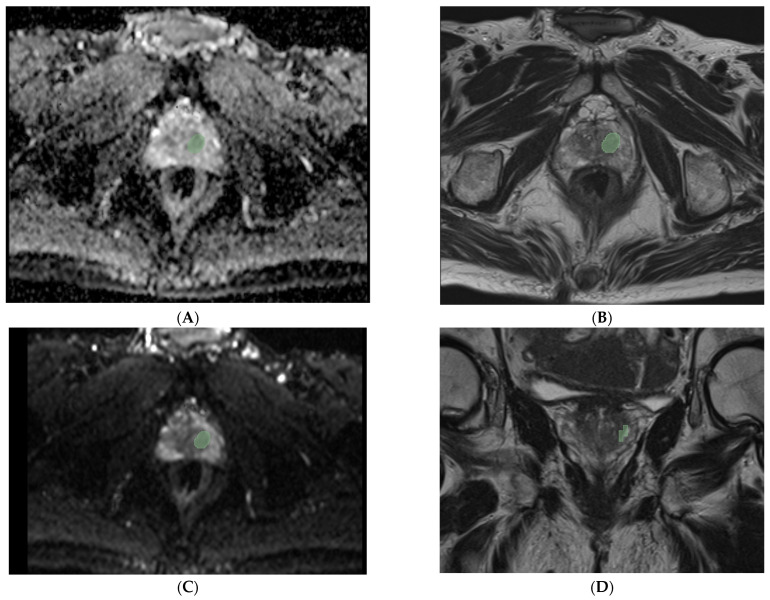
Prostatic nodule (green zone) in ADC map (**A**), axial T2 (**B**), DWI (**C**), and coronal T2 (**D**).

**Table 1 diagnostics-15-00421-t001:** Patients’ characteristics.

Category	Number
Number of patients	95
Patients with metastatic nodule at lymphadenectomy	35
Patients without metastatic nodule at lymphadenectomy	60
Race	Caucasian
Age (Mean)	60.3 (40–80)
PSA [ng/mL] (Median, range)	4.5 (1.0–7.0)
Period of mp-MRI	2016–2023
Gleason grade (Median)	7
Tumor target zone peripheral	63
Tumor target zone transition	32
Disease grade	T3 (36) 37.89% T2 (30) 31.58% T3–5 (29) 30.5%

**Table 2 diagnostics-15-00421-t002:** The confusion matrix reveals the disparity between accuracy and AUC, illustrating how misclassification of the minority class can affect the model’s overall effectiveness.

Accuracy	Precision (Class 0)	Precision (Class 1)
**84%**	80%	88%
**87%**	75%	80%

**Table 3 diagnostics-15-00421-t003:** Summary of the performance of the Random Forest model applied for the two groups based on the location inside the prostate gland. PZ indicates the nodules of the peripheral zone. TZ includes the nodules of the transitional zone.

Group	n	Accuracy	Precision (Class 0)	Precision (Class 1)	Recall (Class 0)	Recall (Class 1)	F1-Score (Class 0)	F1-Score (Class 1)
**PZ**	63	84%	80%	88%	85%	83%	82%	85%
**TZ**	32	87%	75%	80%	77%	78%	76%	79%

**Table 4 diagnostics-15-00421-t004:** The accuracy of RF model for calculating the positive lymph node involvement (PL) and negative lymph node involvement (NL) in peripheral zone (PZ) and transitional zone (TZ).

		Accuracy	AUC
**PZ**	**PL**	80%	75%
	**NL**	85%	83%
**TZ**	**PL**	86%	78%
	**NL**	88%	84%

**Table 5 diagnostics-15-00421-t005:** Top 5 most influential radiomic features in terms of RF importance for positive and negative LN in the PZ.

	Radiomic Features	RF Importance
**Positive LN**	ADCshapeMaimum2DDiameterRow	0.03
	T2_noduloglcmDifferenceVariance	0.02
	T2_noduloshapeElongation	0.02
	ADCglrlmGrayLevelNonUniformity	0.02
	T2_noduloglcmId	0.02
**Negative LN**	DWIshapeMaimum2DDiameterRow	−0.71
	ADCglrlmLongRunHighGrayLevelEmphasis	−0.68
	ADCglcmCorrelation	−0.66
	DWIshapeMaimum3DDiameter	−0.63
	DWIgldmLargeDependenceEmphasis	−0.62

**Table 6 diagnostics-15-00421-t006:** Top 5 most influential radiomic features in terms of RF importance for positive and negative LN in the TZ.

	Radiomic Features	RF Importance
**Positive LN**	DWIglcmDifferenceAverage	0.05
	DWIglcmIdm	0.04
	ADCfirstorderTotalEnergy	0.03
	DWIglrlmGrayLevelNonUniformityNormalized	0.03
	ADCgldmGrayLevelVariance	0.02
**Negative LN**	DWIglcmId	−0.56
	DWIglcmIdmn	−0.53
	DWIglcmIdn	−0.50
	DWIglrlmGrayLevelNonUniformityNormalized	−0.49
	DWIglcmIdm	−0.48

**Table 7 diagnostics-15-00421-t007:** Top 5 most influential radiomic features in terms of RF importance for positive and negative LN in the whole gland.

	Radiomic Features	RF Importance
**Positive LN**	DWIfirstorderTotalEnergy	0.04
	DWIfirstorderEnergy	0.03
	T2_noduloglcmDifferenceAverage	0.02
	T2_noduloglszmLargeAreaLowGrayLevelEmphasis	0.02
	DWIglrlmShortRunLowGrayLevelEmphasis	0.02
**Negative LN**	DWIglcmIdmn	−0.45
	DWIgldmLargeDependenceEmphasis	−0.43
	DWIglcmIdn	−0.41
	T2_noduloshapeMinorAisLength	−0.38
	DWIglcmId	−0.35

## Data Availability

All datasets generated for this study are included in the manuscript.

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
