# Peer review of "Promising Results About the Possibility to Identify Prostate Cancer Patients Employing a Random Forest Classifier: A Preliminary Study Preoperative Patients Selection"

_diagnostics, 2025, doi:10.3390/diagnostics15040421_

Round 1
Reviewer 1 Report
Comments and Suggestions for Authors
The manuscript highlights the potential of machine learning (ML) in advancing prostate cancer diagnostics but limits its exploration to a single algorithm—Random Forest (RF). While Random Forest is robust and effective for high-dimensional data, it represents only a fraction of the available ML landscape. The authors do not justify their exclusive use of RF over other algorithms like Support Vector Machines (SVM), Gradient Boosting Machines (e.g., XGBoost), or deep learning approaches such as Convolutional Neural Networks (CNN), etc which are increasingly utilized in radiomics and medical imaging studies. However, I have some suggestion to improve this manuscript:
1) Title:
The current title, "Promising results about the possibility to identify Prostate Cancer Patients employing Machine Learning analysis: a preliminary study preoperative patients selection," suggests a broader exploration of machine learning. This is misleading given the exclusive use of Random Forest.
2) Abstract
a) Please clearly state that the study focuses on the Random Forest model, distinguishing it from broader ML studies.
b) Please include a statement on limitations, such as sample size and retrospective design.
3) Introduction
a. Please elaborate on the advantages of Random Forest in handling high-dimensional datasets.
b. There is an inconsistency in referencing—e.g., citations [2] and [3] are not elaborated.
4) Methods
a. Provide more details on the segmentation process and explain why specific radiomic features were prioritized.
b. The mention of GPT-4 for statistical analysis seems unconventional and unreferenced.
c. The authors should include a detailed rationale for selecting Random Forest, addressing questions such as:
- Why was Random Forest deemed suitable for this dataset?
- Were other ML algorithms considered or compared during preliminary analysis?
5) Results
a. Adding a comparative performance analysis of RF against alternative ML methods
b. Why the reason model performed poorly (such as: Precision and Recall for Class 0 and Class 1, then overfitting could occur if the model heavily relies on a few specific features) Please analyze potential reasons ?
Because actually, with only 95 patients and 35 positive cases of lymph node involvement, the dataset is imbalanced.
6) Discussion:
The discussion should address the limitations of relying solely on Random Forest and highlight the potential benefits of exploring other ML techniques in future research.
- 7) Conclusion:
Please Suggest specific future research, such as how validation in larger, what multi-center cohorts, and so on.
Author Response
The manuscript highlights the potential of machine learning (ML) in advancing prostate cancer diagnostics but limits its exploration to a single algorithm—Random Forest (RF).
While Random Forest is robust and effective for high-dimensional data, it represents only a fraction of the available ML landscape. The authors do not justify their exclusive use of RF over other algorithms like Support Vector Machines (SVM), Gradient Boosting Machines (e.g., XGBoost), or deep learning approaches such as Convolutional Neural Networks (CNN), etc which are increasingly utilized in radiomics and medical imaging studies. However, I have some suggestion to improve this manuscript:
1) Title:
The current title, "Promising results about the possibility to identify Prostate Cancer Patients employing Machine Learning analysis: a preliminary study preoperative patients selection," suggests a broader exploration of machine learning. This is misleading given the exclusive use of Random Forest.
Thanks for the suggestion. We modified the title
2) Abstract
a) Please clearly state that the study focuses on the Random Forest model, distinguishing it from broader ML studies.
b) Please include a statement on limitations, such as sample size and retrospective design.
Thanks for the comment. We added the details in the ABS section as suggested
3) Introduction
a. Please elaborate on the advantages of Random Forest in handling high-dimensional datasets.
Thanks to the Reviewer for the suggestion. A sentence about the RF use advantages has been added “Among different ML alghorytms, Random Forest has demonstrated highly effective for handling high-dimensional datasets due to several advantages such as feature subset selection, dimensionality reduction without explicit feature elimination, resistance to overfitting and parallel processing.”
There is an inconsistency in referencing—e.g., citations [2] and [3] are not elaborated.
Reference 2 was deleted.
4) Methods
a. Provide more details on the segmentation process and explain why specific radiomic features were prioritized.
Thanks to reviewer for the suggestion. In the MM section a paragraph about different kind of used features was added. Evenmore the rationale of the features types choice was added in the two discussion section (“The features selected and previously extracted from MRI sequences (T2W, DWI, ADC), were chosen because they are directly linked to the biological and physiological characteristics of the tumor and surrounding tissues, which are crucial for diagnosing and staging prostate cancer. Including these features provides potentially more useful diagnostic information than other features that may be less relevant to lymph node staging.”).
The mention of GPT-4 for statistical analysis seems unconventional and unreferenced.
A Reference has been added for the use of GPT-4 as indicated by the reviewer
The authors should include a detailed rationale for selecting Random Forest, addressing questions such as:
- Why was Random Forest deemed suitable for this dataset?
- Were other ML algorithms considered or compared during preliminary analysis?
Thank you for this suggestion. In our previously study (Comparison between Three Radiomics Models and Clinical Nomograms for Prediction of Lymph Node Involvement in PCa Patients Combining Clinical and Radiomic Features. Cancers (Basel), Santucci D, Ragone R, Vergantino E, Vaccarino F, Esperto F, Prata F, Scarpa RM, Papalia R, Beomonte Zobel B, Grasso FR, Faiella E. 2024 Jul 31;16(15):2731. doi: 10.3390/cancers16152731. PMID: 39123458; PMCID: PMC11311324) we explored three algorithms (RF, Logistic regression and SVM) in order to predict LNS in PCa patients and we obtained significantly best performance using RF model. For its structure, for the specific aim and also for these previous results we decided to focus our analysis usig RF model. We specified it in the discussion section
Evenmore, we added some sentences about RF model, its use and our choice and the following sentences have been added in the intro and disc sections
“The segmentations were performed using a dedicated tool (“segment editor”), creating a mask of the region of Interest (ROI) on the three different dimensional planes (axial, sagittal and coronal), obtained a Volume of Interest (VOI). As we can see in Fig 1, the ROI is represented by a green zone on the images”
“RF is highly effective for high-dimensional datasets due to its ability to automatically identify relevant features and handle noisy or irrelevant data. Its ensemble approach reduces overfitting, bias, and variance while capturing non-linear relationships without extensive preprocessing. RF is scalable, supports both classification and regression tasks, and handles imbalanced datasets effectively. Its robustness and versatility make it suitable for applications in bioinformatics, finance, and radiomics analysis… In our previous study we explored three algorithms (RF, Logistic regression and SVM) in order to predict LNS in PCa patients and we obtained significantly best performance using RF model. [21]”
5) Results
a. Adding a comparative performance analysis of RF against alternative ML methods
Thank you again for the point. As previously specified, we based the choise of RF model on previous results
Why the reason model performed poorly (such as: Precision and Recall for Class 0 and Class 1, then overfitting could occur if the model heavily relies on a few specific features) Please analyze potential reasons ?
Thanks for the comment. We explained the possible cause of the poor performance in the discussion section. “The model’s poor performance, particularly in terms of precision and recall for Class 0 and Class 1, is likely due to the dataset’s imbalance. With only 95 patients and 35 positive cases of lymph node involvement, the model tends to favor the majority class (Class 0), leading to missed detections (low recall) and false positives (low precision) for Class 1. Additionally, the small dataset increases the risk of overfitting, as the model may rely excessively on specific features that do not generalize well.”
6) Discussion:
The discussion should address the limitations of relying solely on Random Forest and highlight the potential benefits of exploring other ML techniques in future research.
Thanks you for the point. However, we decided consciously to perform a RF model basing to previous results and basing on its intrinsic capability to explore this kind of datasets
- 7) Conclusion:
Please Suggest specificfuture research, such as how validation in larger, what multi-center cohorts, and so on.
Following your suggestion the following paragraph has been included: “Perspective strategies include higher patient sample, multi-cohort study design, possibly with patient differing on clinic-anamnestic characteristics and employing different MR scans (3T vs 1.5T), and deep learning techniques application, and longitudinal studies assessment in order to assess the impact of predictive models on clinical outcomes. Even more validation with larger dataset and open-source sharing would be appreciated in the next future”.
Reviewer 2 Report
Comments and Suggestions for Authors
This study investigates the effectiveness of an AI model in predicting lymph node involvement in prostate cancer using MRI data and radiomic features. The analysis included 95 patients who underwent multiparametric MRI, prostatectomy, and lymphadenectomy between 2016 and 2022. Radiomic features from T2-weighted, DWI, and ADC sequences were processed using a Random Forest approach, alongside clinical data such as PSA levels and Gleason scores. The proposed approach demonstrated high accuracy, particularly in the peripheral and transitional zones, with key predictive features identified from ADC and DWI sequences. The findings suggest that AI-driven radiomic analysis can significantly aid in preoperative lymph node assessment and patient selection, though further validation with larger cohorts is recommended.
In general, the paper is promising but, I have a few important remarks:
- The authors must add a new section that describes similar or related approaches from the literature
- The authors must give more details about the main pipeline of their approach. If I understood correctly, they had a retrospective cohort of patients and their scans, a large set of radiomic features have been extracted from these scans and involved in an ML classifier (a Random Forest approach) for predicting (in a binary way) the presence or the absence of some nodules. Based on the trained RF models, they have selected the most important features and stopped the pipeline. I recommend continuing the pipeline by analyzing if a new ML model, trained only on those selected features, can correctly classify the nodules for patients from both retrospective and prospective studies.
- Please mention the used technique for computing feature importance (e.g. based on Gini impurity, based on permutations, based on SHAP values, etc.)
- Table2 – please give a short explanation about the meaning of classification labels (class 0 and class 1)
- Please indicate if the results reported in Table 2 are obtained by the same RF model for both groups (PZ and TZ) or by two models (one for each group)
- The authors have to give more details about how they used an LLM (GPt-4 or other models) for the statistical analysis of radiomic features and for establishing the top-k most influential radiomic features in both zones (PZ and TZ) – those features mentioned in Tables 4 and 5 – but also in the entire gland (table 6). Please indicate the prediction model that was used in all these 3 scenarios.
- Please give more details about the training setup of all the classification models (the values of various hyper-parameters, if the training was performed in a cross-validation framework or not, what the convergence of the training process is, etc.)
- Please mention if the reported results (in Tables 2,3,4,5,6) correspond to the validation subset or the training subset.
Author Response
This study investigates the effectiveness of an AI model in predicting lymph node involvement in prostate cancer using MRI data and radiomic features. The analysis included 95 patients who underwent multiparametric MRI, prostatectomy, and lymphadenectomy between 2016 and 2022. Radiomic features from T2-weighted, DWI, and ADC sequences were processed using a Random Forest approach, alongside clinical data such as PSA levels and Gleason scores. The proposed approach demonstrated high accuracy, particularly in the peripheral and transitional zones, with key predictive features identified from ADC and DWI sequences. The findings suggest that AI-driven radiomic analysis can significantly aid in preoperative lymph node assessment and patient selection, though further validation with larger cohorts is recommended.
In general, the paper is promising but, I have a few important remarks:
- The authors must add a new section that describes similar or related approaches from the literature
Dear Reviewer, thank you for your suggestion. We have added some paragraphs exploring the poor literature on this topic.
- The authors must give more details about the main pipeline of their approach. If I understood correctly, they had a retrospective cohort of patients and their scans, a large set of radiomic features have been extracted from these scans and involved in an ML classifier (a Random Forest approach) for predicting (in a binary way) the presence or the absence of some nodules. Based on the trained RF models, they have selected the most important features and stopped the pipeline. I recommend continuing the pipeline by analyzing if a new ML model, trained only on those selected features, can correctly classify the nodules for patients from both retrospective and prospective studies.
Thank you for your comment. In our previous study (Comparison between Three Radiomics Models and Clinical Nomograms for Prediction of Lymph Node Involvement in PCa Patients Combining Clinical and Radiomic Features. Cancers (Basel), Santucci D, Ragone R, Vergantino E, Vaccarino F, Esperto F, Prata F, Scarpa RM, Papalia R, Beomonte Zobel B, Grasso FR, Faiella E. 2024 Jul 31;16(15):2731. doi: 10.3390/cancers16152731. PMID: 39123458; PMCID: PMC11311324) we explored three algorithms (RF, Logistic regression and SVM) in order to predict LNS in PCa patients and we obtained significantly best performance using RF model. For its structure, for the specific aim and also for these previous results we decided to focus our analysis usig RF model. We specified it in the discussion section
- Please mention the used technique for computing feature importance (e.g. based on Gini impurity, based on permutations, based on SHAP values, etc.)
Thank you for the clarification.
In the Materials and Methods section, we have included a subsection discussing the Gini Impurity, the significance of the classes and training setup.
- Table2 – please give a short explanation about the meaning of classification labels (class 0 and class 1)
Thanks for the comment. In the Materials and Methods section, we have included a subsection discussing the Gini Impurity, the significance of the classes and training setup.
- Please indicate if the results reported in Table 2 are obtained by the same RF model for both groups (PZ and TZ) or by two models (one for each group)
The same RF model was applied for the two groups. We specified it on the MM section.
- The authors have to give more details about how they used an LLM (GPt-4 or other models) for the statistical analysis of radiomic features and for establishing the top-k most influential radiomic features in both zones (PZ and TZ) – those features mentioned in Tables 4 and 5 – but also in the entire gland (table 6). Please indicate the prediction model that was used in all these 3 scenarios.
Thank you for the clarification.
We have added the following paragraph to the Results section:
"A single RF model was created considering only patients with prostate cancer. Subsequently, the same model was applied exclusively to patients with prostate cancer in the transition zone (TZ) and the peripheral zone (PZ)."
- Please give more details about the training setup of all the classification models (the values of various hyper-parameters, if the training was performed in a cross-validation framework or not, what the convergence of the training process is, etc.)
In the Materials and Methods section, we have included a subsection discussing the Gini Impurity, the significance of the classes and training setup.
Thank you for the clarification.
We have added a paragraph to the MM section.
- Please mention if the reported results (in Tables 2,3,4,5,6) correspond to the validation subset or the training subset.
Thank you for the clarification.. They correspond to the validation subset. We add a senteces about it.
Reviewer 3 Report
Comments and Suggestions for Authors
Overall, the study may be interesting and useful, but the way the results are presented is absolutely unacceptable, the tables are huge and orange, the calculations have 6 decimal places, completely unnecessary. The demographic characteristics of the patients do not indicate the average value, the standard deviation, and the Disease grade does not indicate percentages.
Author Response
Overall, the study may be interesting and useful, but the way the results are presented is absolutely unacceptable, the tables are huge and orange, the calculations have 6 decimal places, completely unnecessary. The demographic characteristics of the patients do not indicate the average value, the standard deviation, and the Disease grade does not indicate percentages.
Thank to the Reviewer for the comments. We modified all the tables and the decimals have been removed. Even more the average values, the SD and the Disease grade were added in the results section and in table 1 and tables were modified as suggested by the reviewer.
Round 2
Reviewer 1 Report
Comments and Suggestions for Authors
The author has thoroughly addressed the revisions.
However, it is recommended to include a confusion matrix of the Random Forest results in the section preceding Table 2 to provide additional clarity and support to the findings.
Author Response
Thank you for the revisions. As suggested, we have added the section before Table 2.

Reviewer 3 Report
Comments and Suggestions for Authors
The publication looks more correct in this version.
Author Response
Thanks for the suggestions.
Best regards
